# ATOMS (Adjustable Trans-Obturator Male System) in Patients with Post-Prostatectomy Incontinence and Previously Treated Urethral Stricture or Bladder Neck Contracture

**DOI:** 10.3390/jcm11164882

**Published:** 2022-08-19

**Authors:** Ane Ullate, Ignacio Arance, Miguel Virseda-Chamorro, Sonia Ruiz, Juliusz Szczesniewski, Carlos Téllez, Fabian Queissert, Juan F. Dorado, Javier C. Angulo

**Affiliations:** 1Clinical Department, Faculty of Biomedical Science, Universidad Europea, Carretera de Toledo, Km 12.500, Getafe, 28905 Madrid, Spain; 2Department of Urology, Hospital Universitario de Getafe, Carretera de Toledo, Km 12.500, Getafe, 28905 Madrid, Spain; 3Department of Urology, Hospital Nacional de Parapléjicos, Carretera de la Peraleda, S/N, 45004 Toledo, Spain; 4Department of Urology and Pediatric Urology, University Hospital Muenster, 48143 Muenster, Germany; 5PeRTICA Statistical Solutions, Av. Leonardo Da Vinci, 8, OF217, Getafe, 28906 Madrid, Spain

**Keywords:** adjustable trans-obturator male system, stress urinary incontinence, artificial urinary sphincter, fixed male sling, sling failure

## Abstract

(1) Background: Male stress incontinence in patients with previously treated urethral or bladder neck stricture is a therapeutic challenge. The efficacy and safety of the adjustable trans-obturator male system (ATOMS) in these patients is unknown. (2) Methods: All patients with primary ATOMS implants in our institution between 2014 and 2021 were included. The outcomes of patients with previously treated urethral or bladder neck stricture (≥6 months before ATOMS implant) and stable 16Ch urethral caliber were compared to those without a history of stricture. The primary endpoint was the dry patient rate, defined as the pad test ≤ 20 mL/day, and complication rate, including device removal. The secondary variable was self-perceived satisfaction using the Patient Global Impression of Improvement (PGI-I) scale. Wilcoxon rank sum test, Fisher’s exact test and logistic regression were performed. (3) Results: One hundred and forty-nine consecutive patients were included, twenty-one (14%) previously treated for urethral or bladder neck stricture (seven urethroplasty, nine internal urethrotomy and five bladder neck incision). After ATOMS adjustment, 38% of the patients with treated stricture were continent compared to 83% of those without (*p* < 0.0001). After weighted matched observations using propensity score pairing, the proportion of continent patients without a previous stricture was 56% (*p* = 0.236). Complications occurred in 29% of the patients with stricture and in 20% of those without (*p* = 0.34). The severity of the complications was distributed evenly among the groups (*p* = 0.42). Regarding self-perceived satisfaction with the implant, 90% of the patients with stricture perceived the results satisfactorily (PGI-I 1–3) compared to 97% of the rest (*p* = 0.167). Stricture was associated with radiotherapy (*p* < 0.0001) and time from prostatectomy to implantation (*p* = 0.012). There was a moderate correlation between previous stricture and the severity of incontinence, both evaluated according to the 24-h pad test (Rho = 0.378; *p* < 0.0001) and the ICIQ-SF questionnaire (Rho = 0.351; *p* < 0.0001). Multivariate analysis for the factors predictive of failure after ATOMS adjustment revealed previous stricture (OR 4.66; 95% CI 1.2–18.87), baseline 24-h pad test (per 100 mL, OR 1.28; 95% CI 1.09–1.52) and final cushion volume (per mL, OR 1.34; 95% CI 1.19–1.55). This model predicted dryness with an AUC of 92%. After the PSMATCH procedure using a propensity score, the model remained unchanged, with the previous stricture (OR 8.05; 95% CI 1.08–110.83), baseline 24-h pad test (per 100 mL, OR 1.53; 95% CI 1.15–2.26) and final cushion volume (per mL, OR 1.45; 95% CI 1.17–2) being independent predictors and an AUC of 93%. (4) Conclusions: ATOMS can be used to treat male stress incontinence in patients with a history of stricture, although the effectiveness of the device is reduced. On the other hand, the security and perceived satisfaction were equivalent for both groups.

## 1. Introduction

Various surgical options are available for the treatment of male stress urinary incontinence (SUI) after radical prostatectomy. The adjustable trans-obturator male system (ATOMS, A.M.I. GmbH, Feldkirch, Austria) is increasingly used for the surgical correction of moderate-to-severe male SUI [1]. Other options such as retro-urethral fixed bulbar slings are preferred for the first-line treatment of mild-to-moderate incontinence after radical prostatectomy in the absence of radiation, and the artificial urinary sphincter (AUS) is probably the best option for severe SUI without residual sphincteric function [2]. However, choosing the best treatment option for a particular patient can be a difficult task [3].

ATOMS provides an efficient alternative for both the first- and second-line treatments after other failed options [4,5,6,7]. This device is placed using a trans-obturator passage and positioned under the urethra in order to achieve ventral compression of the bulbar urethra ventrally. Unlike the fixed sling, it can even be used in patients with a partially damaged sphincter, and that enlarges its range of applications. The mode of action is a mixture of direct compression of the bulbar urethra and, in the case of good urethral elasticity, a distribution of pressure to the membranous urethra [8]. The scrotal port connected to the sub-urethral cushion allows simple adjustments by percutaneous puncture through the scrotal skin, which can be done even years after the device was originally implanted. In general terms, ATOMS achieves better continence rates than other fixed male slings and lower complication rates than the AUS [2,6]. ATOMS can also be used after radiotherapy, although, in these cases, it is not as effective as in patients without radiation [9,10].

Urethral fragility is a very special condition often associated with male SUI. The risk of incontinence after radical prostatectomy is increased if additional surgical interventions are required for urethral or bladder neck stricture, as the external urethral sphincter may be damaged and/or involved in the fibrotic process. On the one hand, urethral erosion, although infrequent, is the most fearful complication of an AUS implant with a compromised urethra and needs reconstruction before a second-line alternative for SUI is considered. Additionally, active urethral or bladder neck stricture contraindicates the use of a device to treat male SUI [11]. As a consequence, the urethral integrity must be achieved and urethral patency needs to be stable before any incontinence device is considered. 

The present study was conducted to investigate the influence of previous urethral and/or bladder neck stricture treatments on patients’ continence, complication rate and self-perceived satisfaction after primary ATOMS implantation.

## 2. Materials and Methods

### 2.1. Study Population

A retrospective cohort multicenter study was undertaken to evaluate the effectiveness, safety and patient reported outcomes (PRO) of a primary ATOMS implant in consecutive patients in a university hospital. Patients with and without previously treated urethral or bladder neck stricture were compared. All patients included had persistent bothersome SUI for more than a year after radical prostatectomy and had been prescribed pelvic floor exercises before ATOMS indication. 

Inclusion criteria for patients with previous treatments of urethral or bladder neck stricture at any location included stable urethral patency with passage of a 16Ch catheter and a minimum 6-month follow-up after treatment of stricture. Cystoscopy ruled out obstruction in every case before ATOMS placement. Patients with urethral repair after AUS or male sling complications also received an ATOMS implant more than 6 months after urethral reconstruction. Radiotherapy, SUI severity and patient age were not limiting factors for inclusion. The study was derived from the current clinical practice. The indication for ATOMS in contrast to another device was made by a physician with the informed consent of the patient. IRB approval was obtained.

The surgical technique for ATOMS placement followed the original description of Seweryn et al. [4,12]. A 14Ch Foley catheter was placed, and delicate urethral dissection without splitting bulbospongiosus muscle was performed, avoiding excessive dissection and with special attention to avoid damaging the urethra. Careful hemostasis was always performed, and drainage was not placed. When necessary, postoperative adjustment was performed in the office starting 2 to 3 weeks after the implantation by the percutaneous injection of physiological sodium chloride solution through the port membrane and thereafter when required at intervals of 4 weeks until either dryness was achieved or the maximum filling capacity of the system was reached.

### 2.2. Study Endpoints

The primary endpoint was the evaluation of effectiveness and safety of ATOMS in patients with post-prostatectomy incontinence and previous history stricture treatment and the comparison with the cohort of patients without stricture. The role of urethral stricture and other variables regarding dryness was evaluated. The secondary endpoint was the evaluation of self-perceived satisfaction in patients with previously treated stricture and without.

### 2.3. Variables Evaluated

The data analyzed included demographics, previous radiotherapy, former history of bladder neck stenosis and urethral stricture, intraoperative and postoperative complications and continence outcomes. The SUI severity of incontinence (pad count, pad test and ICIQ-SF) was registered at the baseline, before ATOMS placement and after ATOMS adjustment. Continence outcomes were evaluated at the time when the adjustment was considered complete. Patients with a 24-h pad test ≤ 20 mL/day were considered dry. Pain was evaluated at discharge by the patient as a visual analog scale (0–10). The number and severity of complications was registered according to Clavien–Dindo classification in the first 3 months after the implant. Late complications including device removal and reasons were also evaluated. The self-assessed Patient Global Impression of Improvement (PGI-I) scale specified: 1 “very much better than before”, 2 “much better”, 3 “better”, 4 “same”, 5 “worse”, 6 “much worse” and 7 “very much worse”. For comparison, the results were pooled as 1–3 vs. the rest. No patient was lost to follow-up.

### 2.4. Statistical Analysis

Statistics were calculated as the median values, interquartile range (IQR), minimum and maximum for continuous variables and as the frequency and percent for categorical data. Differences were calculated by the Wilcoxon test for continuous variables and Fisher’s exact test for categorical variables. A *p*-value < 0.05 was considered significant. Propensity score pairing was used to maximize the between-group comparability at the baseline. Logistic regression was performed using a stepwise model (entry 0.15 and stay criterium 0.1) to evaluate independent variables determinant of dryness after device adjustment. The area under the receiver operating characteristic (ROC) curve for the selected model and different combinations of predictive factors was calculated. The statistical analysis was developed using Statistical Analysis System 9.3 (SAS Institute Inc., Cary, NC, USA).

## 3. Results

One hundred and forty-one consecutive patients with primary ATOMS implant for SUI after prostate cancer treatment in a single institution were included in the study. The device used was the one with a silicone-covered pre-attached scrotal port. Table 1 summarizes the clinical data. The median follow-up after the ATOMS implant was 51 (IQR: 26, range 12–90) months. 

### 3.1. Comparison of Baseline and Operative Data among Populations

Of the 149 patients with primary ATOMS implantation, 21 (14.1%) had been previously treated for urethral or bladder neck stricture. In the remaining 128 patients (85.9%), there was no history of urethral stricture or bladder outlet obstruction. Stable urethral patency was confirmed in all cases before ATOMS device implantation. Stricture recurrence was not detected in any case during follow-up.

The population with previous stricture was very heterogeneous. The location was pendulous in four (19%), bulbar in four (19%), bulbomembranous in eight (38%) and bladder neck in five (24%). The therapeutic approach was also individualized for each case, always before device implantation, and consisted of direct vision internal urethrotomy (DVIU) in nine (43%), urethroplasty in seven (33%) and bladder neck endoscopic incision in five (24%). Appendix A (Appendix A) presents the main characteristics of each patient with a previous stricture.

Among the patients treated with urethroplasty, the surgical technique used was excision and primary anastomosis (EPA) (*n* = 3), double face (dorsal inlay plus ventral buccal mucosa graft onlay) augmentation (*n* = 1) and dorsal buccal mucosa graft (BMG) onlay for long bulbomembranous stricture (*n* = 3). In two patients, EPA was performed at the time of AUS explant due to urethral erosion. In the remaining five cases, urethroplasty was performed for bulbomembranous stricture, either before radical prostatectomy (two cases) or after prostate cancer surgery (three cases). In one case, bulbomembranous stricture was repaired simultaneous to urorectal fistula through a perineal approach. In one of the patients with bladder neck contracture, a robotic Y-V plasty was also performed, and in three cases, urorectal fistula was repaired using the York-Mason technique at a later step, always before ATOMS implant. 

Radiotherapy was more frequent in the population with stricture compared to the rest (42.9 vs. 9.4%, *p* < 0.0001) and, also, the use of androgen deprivation (23.8 vs. 3.1%, *p* = 0.006). Additionally, the time since prostatectomy was higher in patients with urethral stricture than those without (median 72 vs. 47 months; *p* = 0.012). Early postoperative pain at the time of hospital discharge was slightly higher in patients with stricture, but the difference was not significant (median 1 vs. 0, *p* = 0.093). Additionally, the operative time was slightly higher for patients with stricture but not significant (median 60 vs. 55 min, *p* = 0.44). No association was detected with stricture and other baseline variables such as patient age, BMI, ASA score, Charlson comorbidity index, previous incontinence surgery and D’Amico prostate cancer risk group (Table 1). 

Regarding the baseline data, the population with stricture suffered more severe baseline SUI. This was confirmed by evaluation of the pad count (*p* < 0.0001); pad test (*p* < 0.0001) and ICIQ-SF total (*p* < 0.0001) and ICIQ-SF-specific questions 1 (*p* < 0.0001), 2 (*p* = 0.001) and 3 (*p* < 0.0001) before ATOMS implant (Table 1). There was a moderate correlation between the previous stricture and severity of the incontinence, both evaluated according to the 24-h pad test (Rho = 0.378; *p* < 0.0001) and the ICIQ-SF questionnaire (Rho = 0.351; *p* < 0.0001).

### 3.2. Comparison of Effectiveness among Populations

After ATOMS adjustment, 38% of the patients with treated stricture achieved continence (≤20 mL 24-h pad test) compared to 83% of those without (*p* < 0.0001) (Table 1). Regarding self-perceived satisfaction with the implant, 90% of the patients with stricture perceived the result satisfactorily (PGI-I 1–3) compared to 97% of the rest (*p* = 0.167). However, the proportion of patients with the best reported outcomes was much lower in patients with stricture than those without (47.6 vs. 73.4%; *p* = 0.006) (Table 1).

The SIU severity with the implant was also higher in patients with stricture regarding the 24-h pad count (median 1 vs. 0 pads; *p* < 0.0001) and 24-h pad test (median 70 vs. 0 mL; *p* < 0.0001) (Figure 1). The final cushion volume (median 17.5 vs. 14.5; *p* = 0.006) and number of fillings (median 3 vs. 1; *p* < 0.0001) were also higher in patients with stricture.

### 3.3. Propensity Score Matching 

After the weighted matched evaluation, 21 patients with previous stricture were weighted matched to 53 patients without stricture. The proportion of patients without stricture achieving dryness was 56%, not statistically different compared to the 38% of those with stricture (*p* = 0.236). The PSMATCHED procedure is shown in Figure 2.

### 3.4. Multivariate Analysis for Factors Predictive of Dryness

A multivariate analysis revealed the previous stricture (OR 4.66; 95% CI 1.2–18.87), baseline 24-h pad test (per 100 mL, OR 1.28; 95% CI 1.09–1.52) and final cushion volume (per mL, OR 1.34; 95% CI 1.19–1.55), which were independent factors predictive of failure; i.e., 24-h pad test ≥ 20 mL after ATOMS adjustment. The area under the curve (AUC) for this predictive model was 92% (Figure 3).

According to this model, the predictive probabilities for a patient to achieve dryness (24-h pad test ≤ 20 mL) can be calculated according to the baseline 24-h pad test and previous history of stricture for different filling volumes of the cushion (Figure 4).

After the weighted matched evaluation differences regarding the baseline pad test and filling volume in the 74 matched observations (21 with stricture and 53 without), they were not significant when using the Satterthwaite method for comparison. The mean ± SD 24-h pad test was 935 ± 232 mL for patients w/o and 955 ± 395 mL for patients with stricture (*p* = 0.839). The mean ± SD filling volume was 20.4 ± 4 mL for patients w/o and 19.6 ± 1.3 mL for patients with stricture (*p* = 0.633). Logistic regression was performed again after using the propensity score, and the model remained unchanged, with an AUC of 93%. The previous stricture (OR 8.05; 95% CI 1.08–110.83), baseline 24-h pad test (per 100 mL, OR 1.53; 95% CI 1.15–2.26) and final cushion volume (per mL, OR 1.45; 95% CI 1.17–2) remained independent predictors in the weighted matched population.

### 3.5. Comparison of Safety Data among Populations

Complications occurred in 29% of the patients with stricture and in 20% of those without (*p* = 0.34). The severity of the complications were distributed evenly among the groups (*p* = 0.46). Similarly, there were no differences between the surgical revision rate, device explant and de novo overactive bladder (Table 1).

## 4. Discussion

Although the functional outcomes after radical prostatectomy are improving, male SUI after prostate cancer treatment remains an important problem. Urethral stricture and bladder neck contracture are among the most common issues that need to be addressed in these patients before incontinence device placement is considered [11]. AUS implantation is the gold standard treatment for severe SUI, although other devices such as ATOMS are increasingly used, especially in cases with moderate incontinence.

Several studies have established that a history of pelvic radiation, previous AUS or urethroplasty negatively affects the outcomes in patients with AUS by compromising the urethral tissue [13,14,15]. Similar data regarding ATOMS are more limited, because this surgical alternative is more recent than AUS, and studies with a large number of patients are still limited. Radiation and a higher baseline severity of SUI are the two main factors identified for failure of the device [9,16]. Other factors, such as diabetes, may also be involved [17]. 

The conditions defining a compromised urethra are varied and include advanced age; previous radiotherapy; associated penile prosthesis implantation; former AUS with urethral erosion and/or atrophy; infected ATOMS; urethral damage during male sling placement; previous urethroplasty and former endoscopic manipulation, including bladder neck incision, direct vision internal urethrotomy and urethral stent placement [11,14,18]. As far as we know, no previous study has addressed the influence of ATOMS implant in patients with fragile urethras. This context is of high interest for urethral reconstructive surgeons, because ATOMS can be an attractive option to avoid AUS placement, especially in patients with one or several previously failed AUS.

We have recent urodynamic evidence that ATOMS placement does not cause obstruction itself during the voiding phase [19,20], but there is no specific study addressing the role of previous solved obstruction in patients receiving ATOMS. On the one hand, despite patients with stricture being at risk of disease recurrence that could impact the long-term functional results, we did not detect a case of stricture recurrence after a median follow-up of 51 months, provided stable urethral patency for at least 6 months was confirmed before ATOMS implant. On the other hand, although ATOMS placement does not anatomically dissect the bulbar urethra circumferentially or separates the bulbospongiosus muscle, some risk of devascularization or atrophy caused by sustained compression remains. In this regard, cases requiring late surgical revision have shown that ATOMS caused less urethral atrophy than AUS [7]. 

Studies addressing the impact of a previous urethroplasty on the outcomes of an AUS are limited both by the retrospective design and sample size. Additionally, most of these studies use different surgical alternatives, such as transcorporal AUS or distal bulbar double cuff placement, which can be another confounding factor for variable results [21,22,23,24]. Inasmuch, radiation and diabetes can be confounding factors often associated with previous urethroplasty [25]. A recent prospective evaluation of the impact of previous urethroplasty on the outcome after AUS implant has recently shown that the explant rate of AUS notably increases in these patients (OR 4.18) [24]. Other authors have also confirmed the increased rate of AUS explant after BMG urethroplasty [25] and compromised urethras for different reasons, including radiation and reintervention [14].

As far as we know, apart from AUS, there is no accumulated experience regarding the role of other incontinence systems and, more specifically, with retro-urethral sling and Pro-ACT in patients with SUI and previously treated urethral or bladder neck stricture. Our experience demonstrates that, although the effectiveness of ATOMS may be reduced in those patients, it can still be effective to control leakage and improve their quality of life.

Full continence was achieved in 38% of these patients, and postoperative SIU severity in cases with persistent incontinence was severely reduced, with a median 1 (IQR 2) pad count and 70 (IQR 80) mL pad test after ATOMS adjustment. Still, though, the effectiveness in this population is significantly lower than in patients without stricture and also compared to the effectiveness reported in the largest multicenter series of ATOMS [9,16,26]. However, when propensity score matching is performed and patients with a previous urethral stricture are weighted adjusted to other factors than the previous urethral stricture, the proportion of full continence achieved is not that different in the 56% range. 

These findings support the assumption that compromised urethral tissue due to several reasons sets patients at a higher risk of failure of any continence device. Urethral mobilization, transection, blood vessel ligation and scarring of tissue may lead to decreased perfusion, which will surely have an impact on the results. This negative influence will most certainly be more deleterious in the cohort of patients with more severe sphincteric damage and also with previous radiation as well.

The logistic regression analysis we show gives interesting data to predict the results in these patients considering mainly the baseline pad test. Additionally, we found the final filling volume of the ATOMS cushion is a surrogate data of different variables that imply a worse prognosis in these patients that need multiple postoperative fillings for device adjustment. The worse results of patients with previously treated stricture can be explained in part because this population represents a subset of patients with more severe sphincteric damage and more severe basal incontinence (using different tools such as a pad test, pad count and ICIQ-SF) and also associated with the previous use of radiotherapy. However, a multivariate analysis revealed a former history of stricture as an independent predictor itself of not achieving dryness, regarding both all the observations and weighted matched observations using propensity score matching.

Despite the reduced effectiveness in this population, the reported outcomes with a patient self-assessed PGI-I rating are very encouraging. Almost half (48%) of this population reported PGI-I = 1 (very much better than before), and 90% reported PGI-I = 1–3 (better than before). Additionally, the revision rate, explant rate and postoperative complications in patients with a previous treatment of urethral stricture could be considered equivalent to the general population. This fact further supports the observation that ATOMS is an attractive option for reconstructive surgeons to consider in a complex patient with SUI after radical prostatectomy, i.e., with a fragile urethra, radiation and previous surgeries. However, no direct comparison between ATOMS and AUS has been undertaken to date [2], thus complicating the decision to recommend one technique in a patient with SUI after prostate cancer and a fragile urethra over another.

Our study has several important limitations. First, this single-center data have been prospectively recorded but have no comparison arm. Secondly, the target population of patients with ATOMS and a previously treated stricture is very heterogeneous with regards not only to the previous surgical approach but also the intrinsic nature of the stricture. Repeated treatments for the stricture, stricture length, anatomical location and time after urethral healing were not considered due to the small number of patients in the cohort and surely would have an impact on the results. Strictly speaking, a bladder neck contracture and a stricture in pendulous urethra have nothing in common. Despite the limitations, the findings presented are encouraging to furtherly analyze the issue in multicenter studies with a larger number of patients and better control of the confounding variables.

## 5. Conclusions

ATOMS can be used to treat male stress incontinence in patients with a history of urethral or bladder neck stricture, although the effectiveness of the device is severely reduced compared to patients without a previous history of stricture. This fact is explained in part because patients with treated stricture suffered from more severe basal incontinence, but a multivariate analysis revealed that a former history of stricture is an independent predictor of not achieving dryness. On the other hand, the security and perceived satisfaction for an ATOMS implant in patients with a previous history of stricture were equivalent to those without stricture. ATOMS is a very interesting option for reconstructive urologists facing SUI after radical prostatectomy in patients with a fragile urethra. Our experience can be useful to counsel this peculiar population of patients.

## Figures and Tables

**Figure 1 jcm-11-04882-f001:**
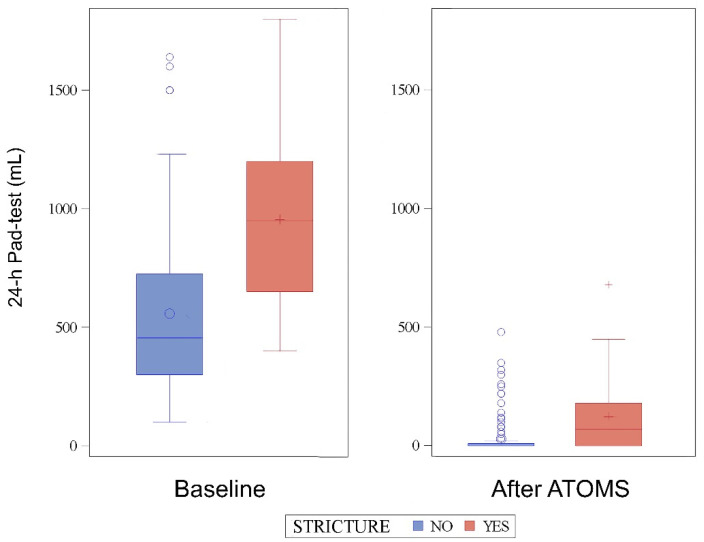
The 24-h pad test evolution before and after the ATOMS implant.

**Figure 2 jcm-11-04882-f002:**
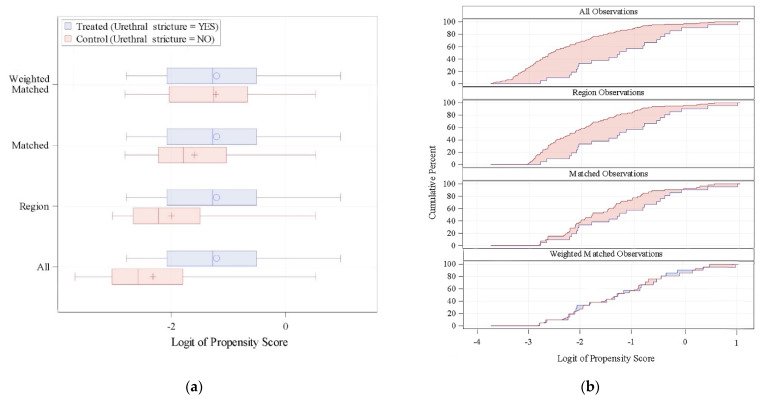
(**a**) Distribution of the Logit of Propensity Scores (LPS). (**b**) Cumulative distribution of the LPS.

**Figure 3 jcm-11-04882-f003:**
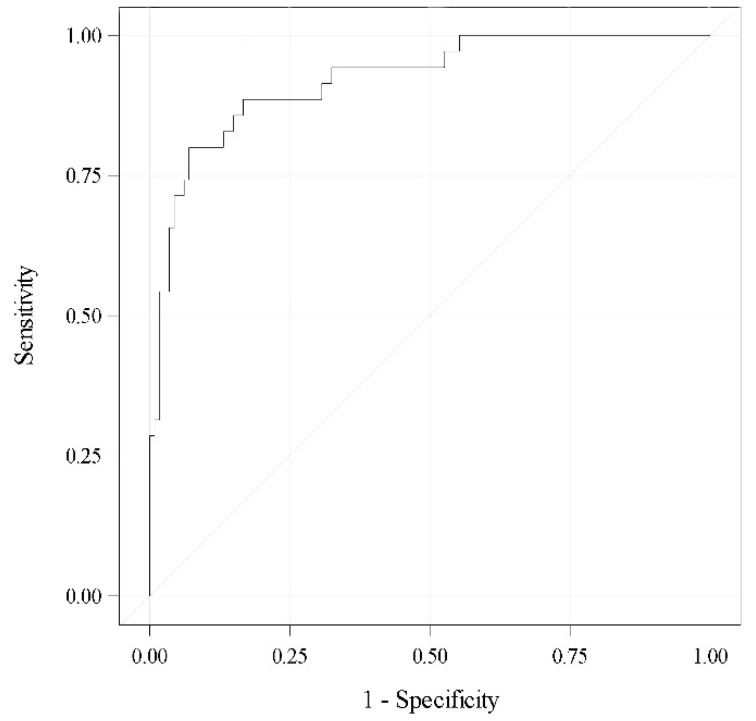
ROC curve for the selected model (AUC 0.92).

**Figure 4 jcm-11-04882-f004:**
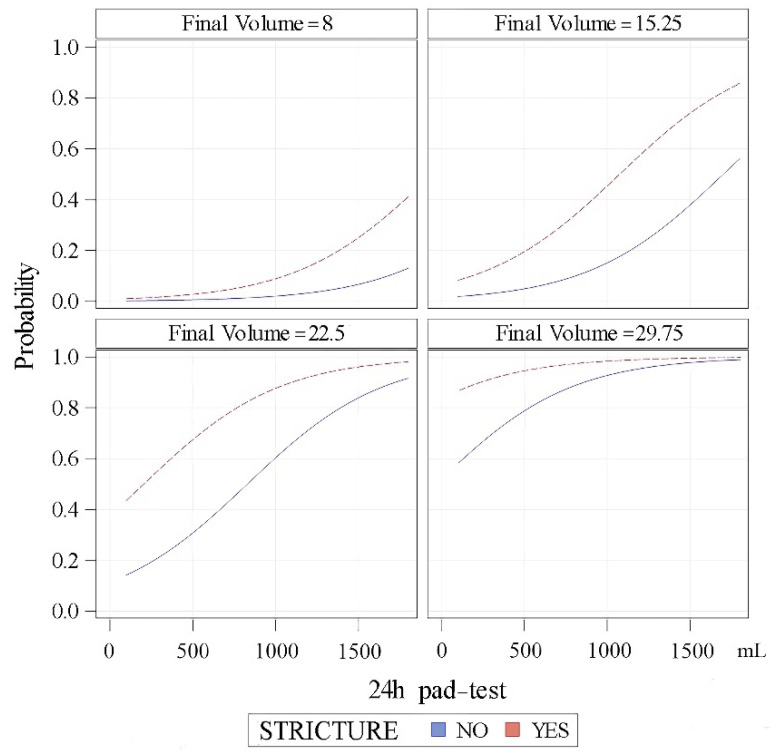
Predictive probabilities of patients achieving dryness (24-h pad test ≥ 20 mL after adjustment) regarding stricture history, 24-h pad test and filling volume.

**Table 1 jcm-11-04882-t001:** Preoperative, operative and postoperative data of the patients included in the study (*n* = 141).

Variable	Total Series (*n* = 149)	With Stricture (*n* = 21)	W/o Stricture (*n* = 128)	*p*-Value
Preoperative data				
Age, years, median (IQR)	70 (7)	69 (8)	70 (7)	0.99
Body mass index, median (IQR, range)	26.6 (4.8)	27 (4.1)	26.3 (4.8)	0.496
ASA score I, *n* (%)	27 (18.1)	4 (19)	23 (18)	0.939
ASA score II, *n* (%)	97 (65.1)	13 (61,9)	84 (65.6)	
ASA score III, *n* (%)	25 (16.8)	4 (19)	21 (16.4)	
Charlson comorbidity index, median (IQR)	4 (2)	4 (2)	4 (2)	0.619
Previous urethral sling surgery, *n* (%)	12 (8.8)	2 (9.5)	10 (7.8)	0.789
Previous radiation, *n* (%)	21 (14.9)	9 (42.9)	12 (9.4)	<0.0001
Androgen deprivation, *n* (%)	9 (6)	5 (23.8)	4 (3.1)	0.003
D’Amico low risk group, *n* (%)	16 (10.7)	2 (9.5)	14 (10.9)	0.319
D’Amico intermediate risk group, *n* (%)	32 (21.5)	2 (9.5)	30 (23.5)	
D’Amico high risk group, *n* (%)	101 (67.8)	17 (81)	84 (65.6)	
Months since prostatectomy, median (IQR)	48 (41)	72 (38)	47 (40)	0.012
24-h pad count (PPD), *n*, median (IQR)	5 (3)	6 (1)	4 (3)	<0.0001
24-h pad test, mL, median (IQR)	500 (460)	950 (550)	455 (425)	<0.0001
ICIQ-SF total, median (IQR, range)	15 (5)	19 (5)	14 (5)	<0.0001
ICIQ-SF Question 1	4 (0)	5 (1)	4 (0)	<0.0001
ICIQ-SF Question 2	4 (2)	6 (2)	4 (2)	0.001
ICIQ-SF Question 3	6 (3)	8 (2)	6 (3)	<0.0001
Operative data				
Operative time, min, median (IQR)	55 (22)	60 (25)	55 (20)	0.44
Perioperative complication, *n* (%)	2 (1.3)	0 (0)	2 (1.6)	1
VAS for pain (0–10), median (IQR) ^(1)^	0 (1)	1 (2)	0 (1)	0.093
Postoperative data				
Postoperative complications ^(2)^, any grade, *n* (%)	31 (22)	6 (28.6)	25 (19.5)	0.46
Grade I ^(2)^, *n* (%)	23 (16.3)	4 (19)	19 (14.8)	
Grade II ^(2)^, *n* (%)	2 (1.4)	1 (4.8)	1 (0.8)	
Grade III ^(2)^, *n* (%)	6 (4.3)	1 (4.8)	5 (3.9)	
Surgical revision, *n* (%)	9 (6)	2 (9.5)	7 (5.8)	0.47
Device explant, *n* (%)	7 (4.7)	2 (9.5)	5 (3.9)	0.26
De novo OAB symptoms, *n* (%)	6 (4)	1 (4.8)	5 (3.9)	0.85
Total filling volume, mL, median (IQR)	15 (8)	17 (5.5)	14.5 (7)	0.006
Number of fillings, median (IQR)	1 (3)	3 (1)	1 (2)	<0.0001
Patients with pad test ≤ 20 mL, *n* (%)	114 (76.5)	8 (38)	106 (82.8)	<0.0001
24-h pad count (PPD) ^(1)^, *n*, median (IQR)	0 (1)	1 (2)	0 (0)	<0.0001
24-h pad test, mL, median (IQR)	0 (15)	70 (180)	0 (10)	<0.0001
PGI-I = 1 (very much better), *n* (%)	104 (69.8)	10 (47.6)	94 (73.4)	0.006
PGI-I = 2 (much better), *n* (%)	26 (17.5)	4 (19)	22 (17.2)	
PGI-I = 3 (better), *n* (%)	13 (8.7)	5 (23.8)	8 (6.2)	
PGI-I = 4 (same), *n* (%)	5 (3.4)	1 (4.8)	4 (3.1)	
PGI-I = 5 (worse), *n* (%)	1 (0.7)	1 (4.8)	0 (0)	

^(1)^ Pain evaluated at discharge, usually on day 1 after surgery. ^(2)^ According to Clavien–Dindo classification. IQR, Interquartile range; ASA, American Society of Anesthesiologists; PPD, pads per day; ICIQ-SF, International Consultation on Incontinence Questionnaire-Short Form; VAS, Visual Analog Scale; OAB, overactive bladder; PGI-I, Patient Global Impression of Improvement.

## Data Availability

Full data will be available upon reasonable request to the corresponding author.

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
