# Peer review of "ATOMS (Adjustable Trans-Obturator Male System) in Patients with Post-Prostatectomy Incontinence and Previously Treated Urethral Stricture or Bladder Neck Contracture"

_jcm, 2022, doi:10.3390/jcm11164882_

Round 1

Reviewer 1 Report

A manuscript provides more detailed description of using Adjustable Transobturator Male System – ATOMS, demonstrating that this system can be effective in patients with previously treated urethral or bladder neck stricture and stable urethral patency at least 6 months after stricture treatment.

The manuscript compares ATOMS to an artificial urinary sphincter (AUS) for stress urinary incontinence (SUI).

In my opinion, the manuscript lacks other treatments for SUI such as the ProACT system, AdVance and AdVanceXP slings. These commonly used treatments are not mentioned in the introduction and discussion sections.

However, the results of this manuscript showed that ATOMS provides the safety and perceived satisfaction of the ATOMS implant in patients with a prior history of strictures was equivalent to those without a stricture. Therefore, the publication of this article is necessary to improve the quality of life of men with a very nasty problem.

Author Response

Manuscript ID: jcm-1822672 RESPONSE TO REVIEWER #1

Comment #1: A manuscript provides more detailed description of using Adjustable Transobturator Male System – ATOMS, demonstrating that this system can be effective in patients with previously treated urethral or bladder neck stricture and stable urethral patency at least 6 months after stricture treatment.

Response to comment #1: Thank you for the comment. Yes, this is the main message of the manuscript.

Comment #2: The manuscript compares ATOMS to an artificial urinary sphincter (AUS) for stress urinary incontinence (SUI). In my opinion, the manuscript lacks other treatments for SUI such as the ProACT system, AdVance and AdVanceXP slings. These commonly used treatments are not mentioned in the introduction and discussion sections.

Response to comment #2: Thank you for the comment. We do not intend to compare systems, but to describe the experience with ATOMS in the special situation of stress urinary incontinence with previously treated urethral or bladder neck stricture. Following the reviewers comment we include a paragraph in the discussion section (sixth pearagraph) regarding the lack of specific experience in this situation with other incontinence systems, such as retro-urethral sling and ProACT compressing system.

Comment #3: However, the results of this manuscript showed that ATOMS provides the safety and perceived satisfaction of the ATOMS implant in patients with a prior history of strictures was equivalent to those without a stricture. Therefore, the publication of this article is necessary to improve the quality of life of men with a very nasty problem.

Response to comment #3: Thank you very much for the appreciation.

Reviewer 2 Report

Whenever a man experiences incontinence because of a therapeutic procedure for obstructive LUTS, that is always bad news and represents a challenge for any urologist. For this type of incontinence, there are several surgical options available, and ATOMS has a very good track record of success.

The manuscript presents the results of a retrospective multicenter study with primary ATOMS implants for patients who have previously undergone treatment for urethral and/or bladder neck stricture. The study design is excellent, with an impressive number of participants and a solid statistical analysis.

An overview of this procedure is presented in the Introduction, which provides essential information on this type of procedure. Materials and methods are presented clearly and understandably. Additionally, the results are reproducible, easy to follow and supported by tables and charts.

There is a strong connection between the Discussions and the literature, and the idea behind the study is explained exquisitely. The study, limitations are described here.

Furthermore, the conclusions reflect the study's main point concisely.

Lastly, I would like to congratulate the authors for their effort and recommend this article for publication because it adds value to the literature and could be helpful in clinical practice.

Author Response

Manuscript ID: jcm-1822672 RESPONSE TO REVIEWER #2

Comment #1: The manuscript from Ullate et al aims to investigate the efficacy of ATOMS device in treating post prostatectomy incontinence in patients who have been previously treated for urethral stricture. It’s a well-designed and performed single center retrospective study without any methodological errors to be recognized.

Response to comment #1: Thank you very much for the comment.

Comment #2: No language issues were recognized that would need editing and the flow is good, thus making it an interesting paper to read.

Response to comment #2: Again, thank you very much.

Comment #3: I have only on comment to make. As the authors have mentioned, there is great heterogeneity between the 21 patients that have been treated previously for a urethral stricture, regarding the nature of the stricture and the type of the procedure that was carried out. Although they have used only one group in their analysis, given that the number of patients is small for sub-group analysis. It would be interesting if they could provide some more baseline characteristics regarding each one of these 21 individuals in a separate table (etc type of stricture, type of intervention, previous radiation, severity of incontinence – post ATOMS continence) and the final outcome of the ATOM placement regarding their continence. Despite the fact that we wouldn’t be able to make any correlations it can be an interesting addition.

Response to comment #3: This is an excellent suggestion. A table is now added as Supplementary Material Table S1 and a sentence is omitted in page 5 paragraph 4 for considered redundant.

Comment #4: In conclusion, it is a well written and interesting paper that, despite its limitations, adds to the literature regarding the use of ATOMS in this specific population.

Response to comment #4: Thank you very much for the comment.

English has been revised by a native speaker also.

Reviewer 3 Report

The manuscript from Ullate et al aims to investigate the efficacy of ATOMS device in treating post prostatectomy incontinence in patients who have been previously treated for urethral stricture. It’s a well-designed and performed single center retrospective study without any methodological errors to be recognized. 

No language issues were recognized that would need editing and the flow is good, thus making it an interesting paper to read.

I have only on comment to make. As the authors have mentioned, there is great heterogeneity between the 21 patients that have been treated previously for a urethral stricture, regarding the nature of the stricture and the type of the procedure that was carried out. Although they have used only one group in their analysis, given that the number of patients is small for sub-group analysis,  It would be interesting if they could provide some more baseline characteristics regarding each one of these 21 individuals in a separate table (etc type of stricture, type of intervention, previous radiation, severity of incontinence – post ATOMS continence) and the final outcome of the ATOM placement regarding their continence. Despite the fact that we wouldn’t be able to make any correlations it can be an interesting addition.

In conclusion, it is a well written and interesting paper that, despite its limitations,  adds to the literature regarding the use of ATOMS in this specific population. 

Author Response

(The authors gave the same response as above.)

Reviewer 4 Report

The aim of the study was to evaluate safety and efficacy of ATOMS. Topic is interesting and the study well built. However, there are a few critical aspects that Authors need to review in order to improve overall quality of manuscript.  

- Since there are some preoperative differences between groups, a propensity scored match analysis chould be performed. 

- Strictures after radical prostatectomy and particularly after adjuvant RT is a recidivant disease. Authors should discuss longterm rates of strictures recurrence after ATOMS implant, which could be a difficult situation to treat. 

- Images of device and surgical steps could be included in order to fully explain surgical procedure. 

Author Response

Manuscript ID: jcm-1822672 RESPONSE TO REVIEWER #3

Comment #1: The aim of the study was to evaluate safety and efficacy of ATOMS. Topic is interesting and the study well built. However, there are a few critical aspects that Authors need to review in order to improve overall quality of manuscript. 

Response to comment #1: Thank you very much for the general comment and also for the suggestions that we consider have improved the quality of the manuscript.

Comment #2: Since there are some preoperative differences between groups, a propensity scored match analysis should be performed.

Response to comment #2: This is a crucial aspect. Following the suggestion of the reviewer we have performed a propensity scored match analysis for patients with and without previous urethral stricture. Material and Methods, results and discussion sections are modified regarding this analysis that we consider has improved the manuscript globally. After weighted matched evaluation the proportion of patients achieving dryness with and without stricture is not statistically different. Also, differences regarding baseline pad-test and filling volume (the other independent variables affecting dryness in the logistic regression analysis presented) are not significant using Satterthwaite method for comparison. However, when the logistic regression is performed after the PSMATCHED procedure the factors evaluated (previous stricture, baseline pad-test and ATOSM filling volume) remained independent and significant to predict dryness. This analysis further confirms the role of previous urethral stricture treatment as an adverse factor of prognosis in patients receiving an ATOMS for stress urinary incontinence after prostatectomy.

Comment #3: Strictures after radical prostatectomy and particularly after adjuvant RT is a recidivant disease. Authors should discuss longterm rates of strictures recurrence after ATOMS implant, which could be a difficult situation to treat.

Response to comment #3: We have not documented stricture recurrence in any case in this series with a median follow-up after ATOMS implant of 51 (IQR 26) months. That means stable urethral patency for at least 6 months after stricture treatment appears desirable before performing ATOMS implant in these patients. Two sentences are included in the results and in the discussion sections that highlight this observation. 

Comment #4: Images of device and surgical steps could be included in order to fully explain surgical procedure.

Response to comment #4: Previous articles referenced in our study (References 4, 7, 8 & 12) already show these images. We are grateful for the reviewer’s comment, but we do not consider this suggestion is necessary after the growing evidence of ATOMS.

English has been revised by a native speaker also.